# Regulation of Cell Wall Degradation and Energy Metabolism for Maintaining Shelf Quality of Blueberry by Short-Term 1-Methylcyclopropene Treatment

**Han Yan [1], Rui Wang [1,*], Ning Ji [1], Jiangkuo Li [2], Chao Ma [1], Jiqing Lei [1], Liangjie Ba [1], Guangzhong Wen [3] and Xiaobo Long [3]**

1. College of Food and Pharmaceutical Engineering, Guiyang University, Guiyang 550003, China
2. Tianjin Key Laboratory of Postharvest Physiology and Storage of Agricultural Products, National Engineering and Technology Research Center for Preservation of Agricultural Produce, Tianjin 301699, China
3. Agriculture and Rural Bureau of Majiang County,
   Qiandongnan Miao and Dong Autonomous Prefecture 557600, China
* Correspondence: gyuruiwang@163.com

**Abstract:** In order to study a short-term and efficient technology by 1-methylcyclopropene (1-MCP) in blueberry, the fruit was treated with 0, 0.5, 1 and 3 µL/L 1-MCP for 2 h then stored at $25 \pm 1$ °C with 40–50% relative humidity (RH) for 9 d. The weight loss, decay incidence, respiration rate, firmness, soluble solid content (SSC), titratable acid (TA), Brix-acid ratio (BAR), sensory evaluation, content of cell wall polysaccharide, activities of cell wall composition-related enzymes and energy metabolism in blueberry were determined during shelf life. The results showed that the weight loss, decay incidence and respiration rate were reduced by 3 µL/L 1-MCP treatment. Compared to other groups, the firmness, the content of TA and anthocyanins were maintained in 3 µL/L 1-MCP-treated blueberry. In contrast, the SSC and BAR were lower compared to those untreated. However, the sensory evaluation of "taste" and "aroma" value showed no differences in all fruits. The content of protopectin, cellulose and hemicellulose was higher in 1-MCP-treated blueberry, accompanied by a decrease in polygalacturonase (PG) and pectin methyl esterase (PME) activity. The content of water-soluble pectin (WSP) was lower in 1-MCP-treated blueberry than untreated ones. The activity of phenylalanine ammonia lyase (PAL), peroxidase (POD), cinnamyl alcohol dehydrogenase (CAD) and 4-coumarate-CoA ligase (4CL) was higher in 1-MCP-treated blueberry than the untreated, which induced more serious lignification. The results of energy metabolism also showed that the 1-MCP treatment could ensure sufficient intracellular energy supply. The 3 µL/L 1-MCP treatment could maintain the shelf quality and retard decomposition of cell wall polysaccharide by ensuring sufficient intracellular energy supply and inhibiting cell wall-degrading enzymes activity. Taken together, this study highlighted an efficient and short-term 1-MCP treatment technique.

**Keywords:** blueberry; 1-MCP; cell wall polysaccharide; shelf life; karst fruit

## 1. Introduction

Guizhou province, which is one of the world's three major karst landscape areas, is located in southwest China, the center of east Asia. Karst plateau mountains typically have thin soil layers, sinkholes, steep slopes and low fertility in their soil [1]. However, Shan et al. (2020) reported that long-term cultivation of fruit plantations decreased mineralization and nitrification rates in calcareous soil in the karst region [2], which is conducive to improving the local karst ecosystem. In 2021, the scale of blueberry plantation had reached $1.5 \times 10^4$ hm² in Guizhou province, which produced 69,000 tons of fresh fruit worth over 1.5 billion yuan (the monetary unit of China). It is very important for Guizhou Province to alleviate poverty and revitalize the countryside.

Blueberry (*Vaccinium* spp.) is rich in phenolic acids and anthocyanins, which are natural antioxidant [3,4] and widely cultivated in Guizhou. However, blueberry is susceptible to mechanical damage and perishables because of its thin skin [5]. On the other hand, postharvest blueberry has strong respiration, which accelerates the softening, weight loss and decay [6]. Since fruit firmness is an important indicator in the commercial quality of blueberry, excess softening can result in notable economic loss. Thus, it will be of great value to maintain the firmness of postharvest blueberry, for this can improve fruit quality and storability as well as provide more time for transportation and sales. It is well known that the decrease in firmness is caused by water loss and the change in structure and composition of cell wall [6]. Among them, the polysaccharide (pectin, cellulose, hemicellulose, lignin) metabolism of cell wall is demonstrated to be the primary factor causing fruit softening [7]. In recent years, several technologies have been used to maintain fruit quality and delay softening of blueberry, such as 1-MCP, sodium nitroprusside, acidic electrolyzed oxidizing water dipping and modified atmosphere storage [3,7–9].

An ethylene receptor blocker, 1-MCP can inhibit endogenous ethylene production and the physiological and biochemical reactions of fruit by irreversibly binding with the ethylene receptor [10]. Previous studies showed that 1-MCP could delay softening and inhibit the polysaccharide decomposition of the cell wall in blueberry, pears, apple and plum [11–14]. Ortiz et al. (2018) reported that 1 µL/L 1-MCP treatment for 12 h could alleviate cell wall degradation in blueberry, resulting in a firmness of 16% higher than the control [14]. Moreover, Grozeff et al. (2017) reported that a combination of nitric oxide and 1-MCP (1 µL/L, 12 h) could maintain the firmness and extend storage-life accompanied with ASA and glutathione content in blueberry [15]. Many similar studies had been previously reported [8,16]. Indeed, the application concentration of 1-MCP in previous studies was usually between 0.3 and 1.0 µL/L, and the treatment time was between 12 and 24 h, which made them similar. In practical production, shortening processing time would not only be beneficial for reducing production costs but also provide more time for transportation and sales.

Cell wall polysaccharides include a variety of components, such as pectin, cellulose, hemicellulose and lignin [6]. In postharvest fruit, the pectin and cellulose disintegrated gradually during storage. Furthermore, various abiotic stresses, such as low temperature, high carbon dioxide and physiological damage could also disturb the cellular homeostasis [17]. For blueberry, the softening of fruits is triggered by the accumulation of ethylene during storage and the disassembly of cell wall polysaccharides, and softening has been delayed by 1-MCP treatment, which inhibited activities of cell wall-degrading enzymes, such as PG and PME [12,14]. The deposition of lignin is the main reason for lignification. Lignin is produced by PAL, POD, CAD and 4CL when plants are subjected to biotic/abiotic stress [18,19]. Suo et al. (2018) reported that the risk of lignification increased in 1-MCP-treated 'Hongyang' kiwifruit [20]. However, to the best of our knowledge, we have not found an available report concerning the lignification of blueberry fruit after 1-MCP treatment.

In order to develop a short-term and efficient treatment technique for 1-MCP in blueberry, the present study investigated the effects of short-term treatment with a high concentration of 1-MCP on maintaining shelf quality and delaying softening in blueberry after harvest. The preservation mechanisms of action were explored in relationship to enzyme activities and cell wall metabolism. Moreover, the energy metabolism in postharvest blueberry was also identified by a chromatographic technique.

## 2. Materials and Methods

### 2.1. Blueberry Samples and Treatment

'Britewell' blueberry fruits were picked from a seven-year old orchard in Majiang (107°59′15″ N, 26°49′23″ E), China. Uniform fruits without injuries were harvested at the same maturity stage, in terms of color and size. Fruits were put into trays (125 ± 2 g/tray) and transported to the laboratory within 2 h after harvest. Fruits were randomly divided

into four groups of 3750 g each (30 trays/group). The four groups of blueberries were placed in polypropylene plastic containers (1 m × 1 m × 1 m) (LEYI Inc., Shanghai, China) and fumigated for 2 h at controlled temperature (25 ± 1 °C), as follows: (1) untreated, (2) 0.5 µL/L 1-MCP (SmartFresh[SM] Inc., Wilmington, Delaware, USA), (3) 1 µL/L 1-MCP and (4) 3 µL/L 1-MCP. After treatment, a shelf experiment was carried out for 9 days at 25 ± 1 °C with 40–50% RH. A total of 360 fruits from each group were placed separately for the determination of weight loss, decay incidence and respiration rate. The detection was performed at 0, 3, 5, 7 and 9 d, in turn. Samples were frozen in liquid nitrogen immediately and then stored at −80 °C for the subsequent measurements.

### 2.2. Determination of Weight Loss, Decay Incidence and Respiration Rate

Weight loss (%) ($n = 60$) was calculated as follows: $\frac{(m_0-m)}{m_0} \times 100$, where $m_0$ is the initial weight and $m$ is the final weight of each sample of fruit. Decay percentage was measured by counting the number of fruits decayed in each group. The decayed fruits were considered rotten if they had visible fungal growth or bacterial lesions on their surface.

The respiration rate was calculated for a constant sixty fruits per replication. Sixty fruits were placed in a 1.0 L sealed plastic box at 25 ± 1 °C for 4 h. The $CO_2$ concentration was determined by a gas analyzer (Checkpoint 3 Premium, Mocon Inc., Minneapolis, MN, USA).

### 2.3. Determination of Firmness, SSC, TA and Anthocyanin Content

Fruit firmness ($n = 18$) was measured with a texture analyzer (SMS Inc., London, England). The texture analyzer was fitted with P/2N probe. Fruits were deformed for a distance of 6 mm at a speed of 2 mm s$^{-1}$, and a 0.049 N trigger force was used.

To determine SSC, blueberry juice was extracted from the blueberries ($n = 18$) after grinding and centrifugation (10,000× $g$, 10 min, 4 °C), and its soluble solid content (SSC) was measured using a digital hand-held refractometer (PAL-1, Atago Inc., Tokyo, Japan). Total titratable acid (TA) was determined by a previously described method (Hui et al., 2018) [17]. Then, BAR was calculated by using following formula: BAR = SSC/TA.

For anthocyanin measurement, 10 g of frozen blueberry fruit was ground, and 0.1 g of powder was added to 1 mL methanol (containing 1% HCl) (Aladdin Inc., Shanghai, China). Homogenized samples were centrifuged (10,000× $g$, 10 min, 20 °C) for 10 min. The absorbance was measured at 250 nm and 700 nm. The anthocyanin content (mg/g) was calculated using the method of Ge et al. [3].

### 2.4. Determination of Cell Wall Composition
#### 2.4.1. Cellulose and Hemicelluloses

Cellulose and hemicellulose contents (mg/g) were measured with an assay kit (Norminkoda Biotechnology Co., Ltd., Wuhan, China). The reagents were added in accordance with the manufacturer's instructions, and the absorbance at 620 nm was measured.

#### 2.4.2. Lignin

Lignin content was measured according to the method of Ge et al. (2019), with modifications [3]. Frozen tissues (2.0 g) were weighed and added to 3.0 mL of 95% cold ethanol (pre-cooled in an explosion-proof refrigerator at 4 °C) and then centrifuged (12,000 g, 20 min, 4 °C). The sediment was washed three times with 95% ethanol (Aladdin Inc., Shanghai, China) and rinsed three times with the mixture of ethanol and n-hexane (1:2) (Aladdin Inc., Shanghai, China). The precipitate was collected and dried at 80 °C, then dissolved in 1.0 mL acetic acid (Aladdin Inc., Shanghai, China)and incubated at 80 °C for 30 min; the reaction was stopped by the addition of 1.5 mL 2.0 mol/L NaOH (Aladdin Inc., Shanghai, China). The content of lignin was expressed as $\Delta_{OD}280$/g FW.

### 2.4.3. Pectin Content

The WSP and protopectin content was measured according to the method of Hamauzu et al. (2011) [21]. In principle, galacturonic acid reacted with carbazole to produce an absorption peak at 530 nm, and the absorbance value at 530 nm is positively correlated with WSP and protopectin content.

### 2.5. Enzyme Activity Assays

### 2.5.1. PAL, POD, CAD and 4CL Activity

The activity of PAL was assayed following the procedure as described by Assis et al. [9], with some modifications. The reaction mixture, consisting of 1.0 mL of the supernatant, was mixed with 2 mL of borate buffer (50 mmol L$^{-1}$, pH 8.8) (Aladdin Inc., Shanghai, China), and 1 mL of 1-phenylalanine (20 mmol L$^{-1}$) (Aladdin Inc., Shanghai, China), was incubated at 40 °C for 1 h. The reaction was terminated by heating the reaction mixture in boiling water for 1 min, and absorbance was read at 290 nm by a UV spectrophotometer (Cary 60, Agilent, Santa Clara, California, USA). The PAL activity was expressed as U/mg.

The enzyme activities of 4CL, CAD and POD were measured according to the methods previously described [22,23]. POD, CAD and 4CL activities were expressed as U/mg.

### 2.5.2. PG and PME Activity

PG activity was measured according to a previous method (Ji et al., 2021) [24]. A volume of 0.1 mL supernatant (the enzyme solution was inactivated by a boiling water bath as a control) was mixed with 0.4 mL poly-galacturonic acid (1%, *w/v*; 37 °C) (Aladdin Inc., Shanghai, China), and the mixture was maintained for 30 min at 37 °C. After cooling, 2.5 mL 3,5-dinitrosaliacylic acid (DNS) (Aladdin Inc., Shanghai, China)reagent was added and the absorbance value was measured at 540 nm. PG activity was expressed as U/mg.

PE activity was determined according to a previous method (Ji et al., 2021) [24]. Frozen samples (0.2 g) were homogenized in 1 mL of a 8.8% (*w/v*) NaCl solution (Aladdin Inc., Shanghai, China)in an ice bath. After centrifugation (10,000× *g*, 20 min, 4 °C), the supernatant was maintained at pH 7.5 with 0.1 M NaOH solution (Aladdin Inc., Shanghai, China). The absorbance changes in 1 minute of the reaction mixture containing 1 mL of 0.5% (*w/v*) pectin (Sigma-Aldrich CN Inc., Shanghai, China)), 0.1 mL bromothymol blue (0.01%) (Aladdin Inc., Shanghai, China)and 0.2 mL supernatant were measured and recorded at 620 nm. PE activity was expressed as U/mg.

### 2.6. Determination of ATP, ADP and AMP Content and Energy Charge

ATP, ADP and AMP were measured according to our previous method described by Li et al. (2014). Results were expressed as mg/kg FW. Energy charge (EC) was calculated by using the following formula: EC = (ATP + 1/2ADP)/(ATP + ADP + AMP).

### 2.7. Sensory Evaluation

Sensory evaluation was conducted using a nine-point hedonic scale and performed in a sensory laboratory at Guiyang University. Ten trained and regular blueberry consumers (there are five men and five women, ranging in age from 20 to 25) were asked to evaluate each sample for the sensory attributes of appearance, color, flavor and taste. For scoring criteria, we referred to Nirmal et al. (2020) [25]. Each attribute was scored as follows: 9 = Like extremely; 8 = Like very much; 7 = Like; 6 = Like slightly; 5 = Neither like nor dislike; 4 = Dislike slightly; 3 = Dislike moderately; 2 = Dislike; 1 = Dislike extremely.

### 2.8. Statistical Analysis

The whole experiment was designed following a randomized experimental design with triplicate samples collected on each sampling date. Data were presented as arithmetic means with standard errors. Statistical analysis was carried out using SPSS version 21 (IBM, Armonk, New York, USA). All the tests were performed with a level of significance of 0.05.

## 3. Results

### 3.1. Weight Loss, Decay Incidence and Respiration Rate

#### 3.1.1. Weight Loss

As revealed in Figure 1a, weight loss was increased in untreated and 1-MCP-treated blueberries. We found that the weight losses of untreated, 1 μL/L 1-MCP-treated and 3 μL/L 1-MCP-treated blueberries were not significantly different between 3 d and 5 d but were lower in the 1 μL/L 1-MCP-treated blueberries than in untreated blueberries between 7 d and 9 d ($p < 0.05$). However, weight loss in the 0.5 μL/L 1-MCP-treated blueberries was significantly higher than in the others.

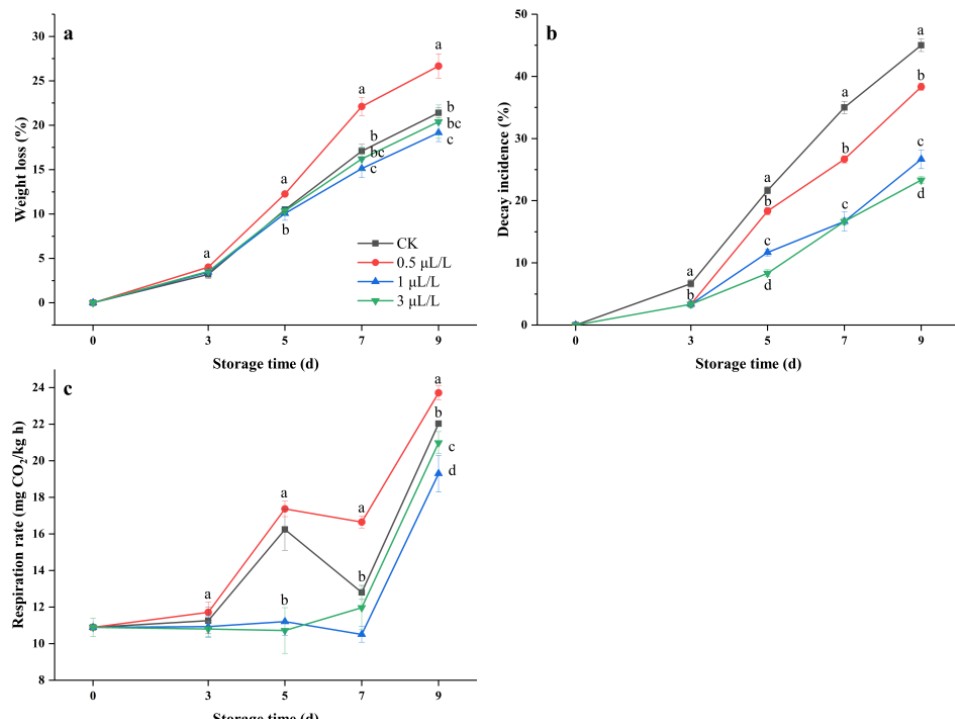

**Figure 1.** Effect of 1-MCP treatment on weight loss (**a**), decay incidence (**b**) and respiration rate (**c**) in blueberries during shelf life at 25 ± 1 °C. Data are the means of three replicates ± standard deviation. Values with different letters are significantly different according to Duncan's test ($p < 0.05$).

The foregoing results indicated that treatment with 1 μL/L 1-MCP suppressed blueberry weight loss in later storage times.

#### 3.1.2. Decay Incidence

As revealed in Figure 1b, the incidence of decay was increased in all groups. However, the rate of decay was slower for 1 μL/L 1-MCP-treated and 3 μL/L 1-MCP-treated blueberries than for the untreated and 0.5 μL/L 1-MCP-treated groups. The decay incidence of untreated blueberries reached 20% in 5 d, while 1 μL/L 1-MCP-treated and 3 μL/L 1-MCP-treated blueberries reached this level only at the end of storage.

Therefore, we found that the decay incidence was effectively suppressed both by 1 μL/L and by 3 μL/L 1-MCP.

#### 3.1.3. Respiration Rate

As revealed in Figure 1c, the respiration rate remained relatively constant both in untreated blueberries and in 0.5 μL/L 1-MCP-treated blueberries from 0 d to 3 d. Similarly, the respiration rate remained relatively constant both in 1 μL/L 1-MCP-treated and in 3 μL/L 1-MCP-treated blueberries from 0 d to 5 d. However, the rate was significantly lower for 1 μL/L 1-MCP-treated and 3 μL/L 1-MCP-treated blueberries than for untreated

blueberries and 0.5 μL/L 1-MCP-treated blueberries at 5 d and 9 d. Hence, according to this observation, the highest respiration rate obviously resulted in weight loss for the 0.5 μL/L 1-MCP-treated group.

### *3.2. Quality Parameters and Anthocyanin Content*
### 3.2.1. Firmness

As revealed in Figure 2a, blueberry firmness was 1.53 N at 0 d and decreased as the storage time increased. For untreated blueberries, firmness rapidly decreased from 0 d to 3 d, then remained relatively constant until 5 d, and finally decreased until the end of storage. Relative to untreated blueberries, 1-MCP treatment delayed the decrease in firmness within a certain range. The firmness dropped to 0.89 N, 0.94 N, 1.20 N and 1.3 N by the end of storage for the untreated, 0.5 μL/L 1-MCP-treated, 1 μL/L 1-MCP-treated and 3 μL/L 1-MCP-treated blueberries, respectively.

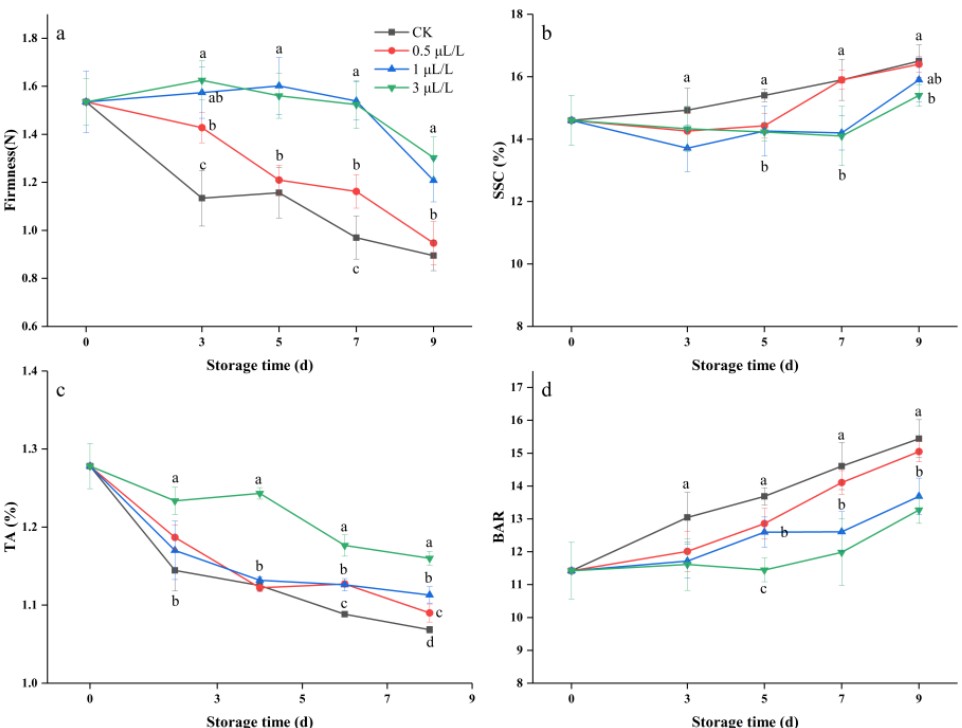

**Figure 2.** Effect of 1-MCP treatment on firmness (**a**), SSC (**b**), TA (**c**) and BAR (**d**) in blueberries during shelf life at 25 ± 1 °C. Data are the means of three replicates ± standard deviation. Values with different letters are significantly different according to Duncan's test ($p < 0.05$).

The foregoing results indicated that 1μL/L1-MCP treatment and 3 μL/L 1-MCP treatment greatly delayed the decrease in firmness in postharvest blueberries.

### 3.2.2. SSC, TA and BAR

As revealed in Figure 2b, blueberry SSC was 14.60% at 0 d and slightly increased as the storage time went on. There were no significant differences between all treatment groups at 3 d. However, we found that blueberry SSC was suppressed by 3 μL/L 1-MCP treatment, which was significantly lower than in untreated blueberries. Blueberry TA content was 1.27% at 0 d, then decreased in all treatment groups until the end of storage. No difference in TA content was found among untreated, 0.5 μL/L 1-MCP-treated and 1 μL/L 1-MCP-treated blueberries, other than at 7 d. In contrast, 3 μL/L 1-MCP treatment greatly delayed the decrease in TA content, which was significantly higher than the other treatment groups over the whole storage time. The BAR was 11.42 at 0 d and increased, as a result of increased SSC and decreased TA, over the whole storage time. The maximum

values were 15.43, 15.04, 13.68 and 13.27 at 9 d for untreated, 0.5 µL/L 1-MCP-treated, 1 µL/L 1-MCP-treated and 3 µL/L 1-MCP-treated blueberry, respectively. There was no difference between untreated and 0.5 µL/L 1-MCP-treated blueberries. In contrast, the BAR of the 3 µL/L 1-MCP-treated blueberries was significantly lower than that of untreated blueberries from 5 d to 9 d ($p < 0.05$).

The foregoing results indicated that the changes in SSC, TA and BAR were delayed by 1-MCP treatment, but 0.5 µL/L 1-MCP treatment was not effective.

### 3.2.3. Anthocyanin Content

As revealed in Figure 3, anthocyanin content was 0.83 mg/g at 0 d. We found that anthocyanin content reached the maximum at 5 d for all treatment groups. However, 1-MCP treatment accelerated the increase in anthocyanin content. The maximum values were 1.37 mg/g, 1.473 mg/g, 1.616 mg/g and 1.844 mg/g for untreated, 0.5 µL/L 1-MCP-treated, 1 µL/L 1-MCP-treated and 3 µL/L 1-MCP-treated blueberries, respectively. The content was significantly higher in 1 µL/L 1-MCP-treated and 3 µL/L 1-MCP-treated blueberries than in the untreated group ($p < 0.05$).

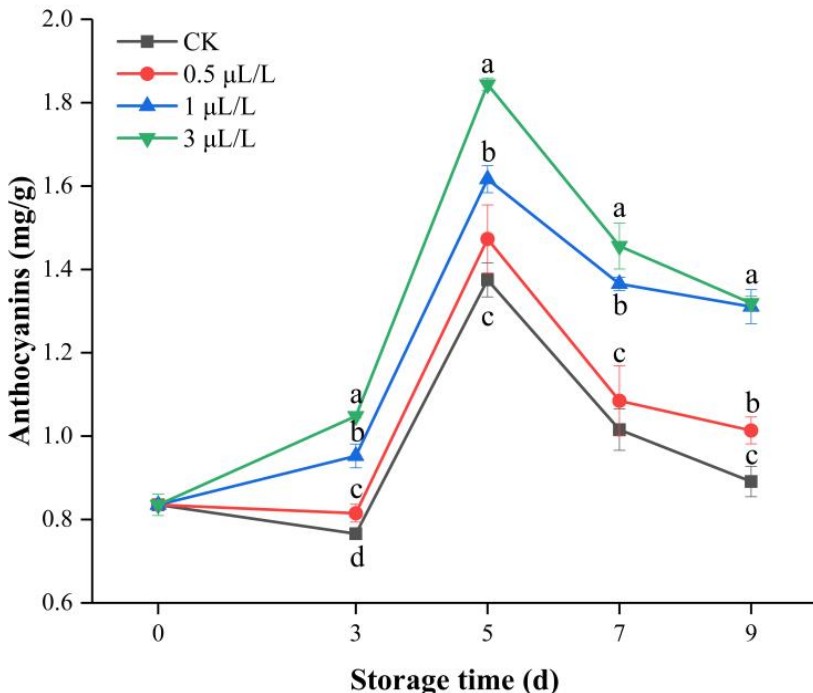

**Figure 3.** Effect of 1-MCP treatment on anthocyanins in blueberries during storage at $25 \pm 1$ °C. Data are the means of three replicates $\pm$ standard deviation. Values with different letters are significantly different according to Duncan's test ($p < 0.05$).

The foregoing results indicated that 1 µL/L 1-MCP and 3 µL/L 1-MCP treatment significantly upregulated the anthocyanin content, whereas the 0.5 µL/L 1-MCP treatment was not effective.

### 3.3. Sensory Evaluation

In order to directly evaluate the effect of 1-MCP treatment on the shelf quality of blueberries, sensory evaluation is necessary. Taking commodity value and consumer acceptability into account, blueberries with a decay incidence not exceeding 20% were chosen to evaluate. As shown in Table 1, the values for firmness, appearance and aroma decreased as the storage time went on. From 5 d to 7 d, the firmness and appearance value was higher in 3 µL/L 1-MCP-treated blueberries than in untreated fruit ($p < 0.05$), while the aroma value showed no difference between 1-MCP-treated and untreated fruit. In

contrast, the taste value of the 1-MCP treated blueberries was slightly increased, while the taste value of untreated blueberries sharply increased at 5 d and was higher than the 1-MCP-treated fruits.

**Table 1.** Sensory evaluation of blueberry during shelf life at 25 $\pm$ 1 $^{\circ}$C.

| Shelf Time | Treatment | Sensory Attribute | | | |
| --- | --- | --- | --- | --- | --- |
| | | Firmness | Taste | Appearance | Aroma |
| 0 d | untreated | 6.8 $\pm$ 1.09 | 6.6 $\pm$ 0.89 | 6.4 $\pm$ 0.89 | 6.4 $\pm$ 0.54 |
| 3 d | untreated | 6.8 $\pm$ 0.83 [a] | 7.0 $\pm$ 0.7 [a] | 5.8 $\pm$ 0.83 [a] | 7.0 $\pm$ 1.64 [a] |
| | 0.5 µL/L | 7.0 $\pm$ 0.70 [a] | 6.4 $\pm$ 0.89 [a] | 6.0 $\pm$ 0.70 [a] | 5.4 $\pm$ 0.89 [a] |
| | 1 µL/L | 6.8 $\pm$ 0.83 [a] | 6.2 $\pm$ 0.98 [a] | 6.6 $\pm$ 0.54 [a] | 5.8 $\pm$ 0.83 [a] |
| | 3 µL/L | 6.6 $\pm$ 0.74 [a] | 5.4 $\pm$ 0.78 [a] | 6.8 $\pm$ 0.83 [a] | 6.0 $\pm$ 0.83 [a] |
| 5 d | untreated | 3.4 $\pm$ 0.83 [b] | 7.2 $\pm$ 0.44 [a] | 4.4 $\pm$ 0.89 [b] | 5.4 $\pm$ 0.98 [a] |
| | 0.5 µL/L | 3.8 $\pm$ 0.83 [b] | 6.5 $\pm$ 0.44 [ab] | 4.6 $\pm$ 0.54 [b] | 4.8 $\pm$ 0.83 [a] |
| | 1 µL/L | 6.4 $\pm$ 0.70 [a] | 6.0 $\pm$ 0.54 [ab] | 5.6 $\pm$ 0.89 [ab] | 5.4 $\pm$ 0.54 [a] |
| | 3 µL/L | 7.0 $\pm$ 0.70 [a] | 5.6 $\pm$ 0.54 [b] | 6.6 $\pm$ 0.54 [a] | 5.2 $\pm$ 0.83 [a] |
| 7 d | untreated | - | - | - | - |
| | 0.5 µL/L | - | - | - | - |
| | 1 µL/L | 6.0 $\pm$ 0.71 [a] | 6.2 $\pm$ 0.83 [a] | 5.2 $\pm$ 0.44 [a] | 6.0 $\pm$ 0.70 [a] |
| | 3 µL/L | 6.6 $\pm$ 0.54 [a] | 5.6 $\pm$ 0.89 [a] | 6.2 $\pm$ 0.83 [a] | 5.8 $\pm$ 0.44 [a] |

Data are the means of nine replicates $\pm$ standard deviation. In columns, different superscript letters at the same shelf time indicate a significant difference according to Duncan's test ($p < 0.05$).

### 3.4. Cell Wall Metabolism and Enzyme Activity

3.4.1. WSP, Protopectin Content and PG and PE Activity

As revealed in Figure 4a, the WSP content in untreated blueberries increased from 0 d to 3 d, then slightly decreased on day 5, and then increased again until the end of storage. A similar trend was observed for the 1-MCP-treated blueberries. The maximum values were 22.31 mg/g, 19.42 mg/g, 14.89 mg/g and 15.06 mg/g for untreated, 0.5 µL/L 1-MCP-treated, 1 µL/L 1-MCP-treated and 3 µL/L 1-MCP-treated blueberry, respectively. The content was significantly higher for 1 µL/L 1-MCP-treated and 3 µL/L 1-MCP-treated blueberries than for untreated and 0.5 µL/L 1-MCP-treated blueberries over the entire storage time ($p < 0.05$). In contrast, protopectin content decreased until the end of storage for all treatments (Figure 4b). For untreated blueberries, it rapidly decreased from 0 d to 5 d then slightly decreased until the end of storage. A similar trend was observed in the 1 µL/L 1-MCP-treated and 3 µL/L1-MCP-treated blueberries. The 3 µL/L 1-MCP treatment significantly suppressed protopectin degradation, which was 2.09 times, 1.62 times and 1.85 times higher than in the untreated blueberries at 5 d, 7 d and 9 d, respectively.

Figure 4c showed that PG activity increased from 0 to 3 d and 5 to 9 d, but decreased at the other storage times regardless of 1-MCP treatment. However, it was significantly lower for the 1 µL/L 1-MCP-treated and 3 µL/L 1-MCP-treated blueberries than for untreated fruit over the whole storage time. The different trend was observed for PE activity, which decreased rapidly at first, then slowly. The PE activity of 1-MCP-treated blueberries was significantly lower than untreated at 3 d and 5 d. In addition, the 3 µL/L1-MCP treatment was most effective in this regard; the PE activity of this group was 45.54% and 30.31% lower than untreated blueberries at 3 d and 5 d, respectively.

The foregoing results indicated that the changes in WSP and protopectin content were maintained by 3 µL/L1-MCP treatment and accompanied by suppressed PG and PE activity.

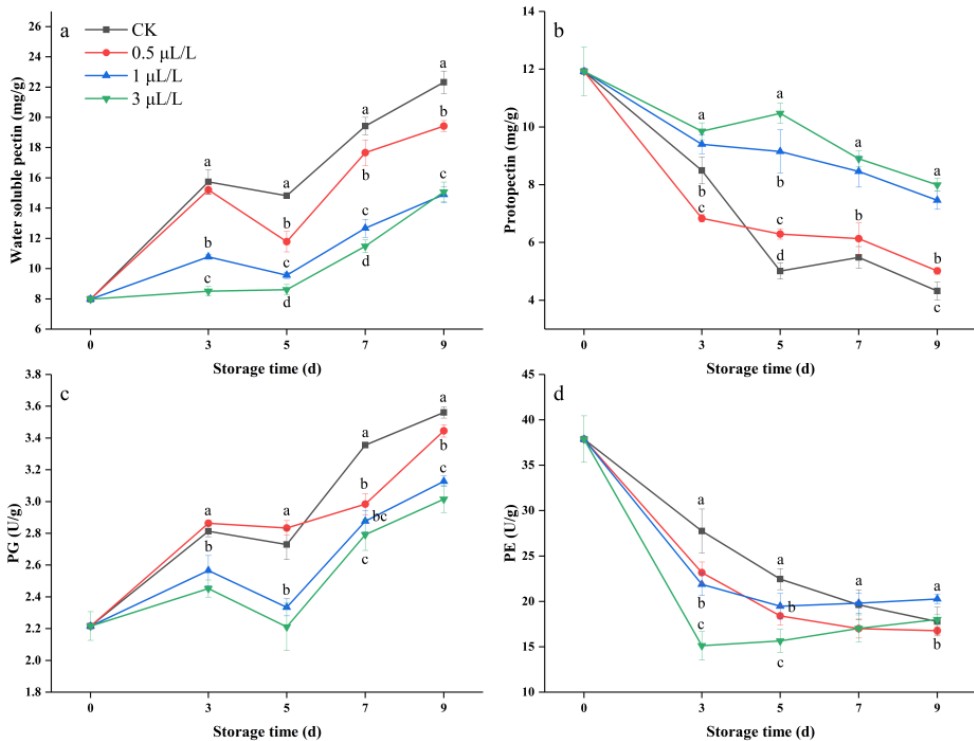

**Figure 4.** Effect of 1-MCP treatment on WSP (**a**), protopectin (**b**), PG activity (**c**) and PE activity (**d**) in blueberries during shelf life at $25 \pm 1$ °C. Data are the means of three replicates $\pm$ standard deviation. Values with different letters are significantly different according to Duncan's test ($p < 0.05$).

### 3.4.2. Lignin Content and PAL, POD, CAD and 4CL Enzyme Activity

As revealed in Figure 5, lignin content was 0.27% at 0 d. For untreated blueberries, the lignin content increased from 0 d to 3 d, then decreased until the end of storage; the maximum was 0.98%. The changes in lignin content for 1 μL/L 1-MCP-treated and 3 μL/L 1-MCP-treated blueberries were similar to untreated blueberries. The maximum values were 1.34%, 1.51% and 2.08% for 0.5 μL/L 1-MCP-treated, 1 μL/L 1-MCP-treated and 3 μL/L 1-MCP-treated blueberries; these values were 1.36 times, 1.54 times and 2.12 times higher, respectively, than thosef for untreated blueberries.

As revealed in Figure 6a, PAL activity increased with storage time. However, it was 1.24 times, 1.21 times, 1.15 times and 1.21 times higher for the 3 μL/L 1-MCP-treated blueberries than for untreated fruit at 3 d, 5 d, 7 d and 9 d ($p < 0.05$), respectively. CAD and 4CL activity had a similar trend to PAL activity (Figure 4b,d). CAD activity was 2.91 times, 3.67 times, 2.41 times and 1.67 times higher for 3 μL/L 1-MCP-treated blueberries than for untreated blueberries at 3 d, 5 d, 7 d and 9 d ($p < 0.05$), respectively. 4CL activity was 1.52 times, 1.54 times, 1.22 times and 1.16 times higher for 3 μL/L 1-MCP-treated blueberry than for untreated at 3 d, 5 d, 7 d and 9 d ($p < 0.05$), respectively. The POD activity increased with storage time in untreated blueberries. However, there was a decrease from 0 d to 3 d, then an increase until the end of storage. POD activity was significantly higher for untreated blueberries than 1-MCP-treated blueberries at 3 d, but lower at 7 d ($p < 0.05$). The change in 3 μL/L 1-MCP-treated blueberries was most obvious and 1.41 times and 1.28 times higher, respectively, than that of untreated blueberries.

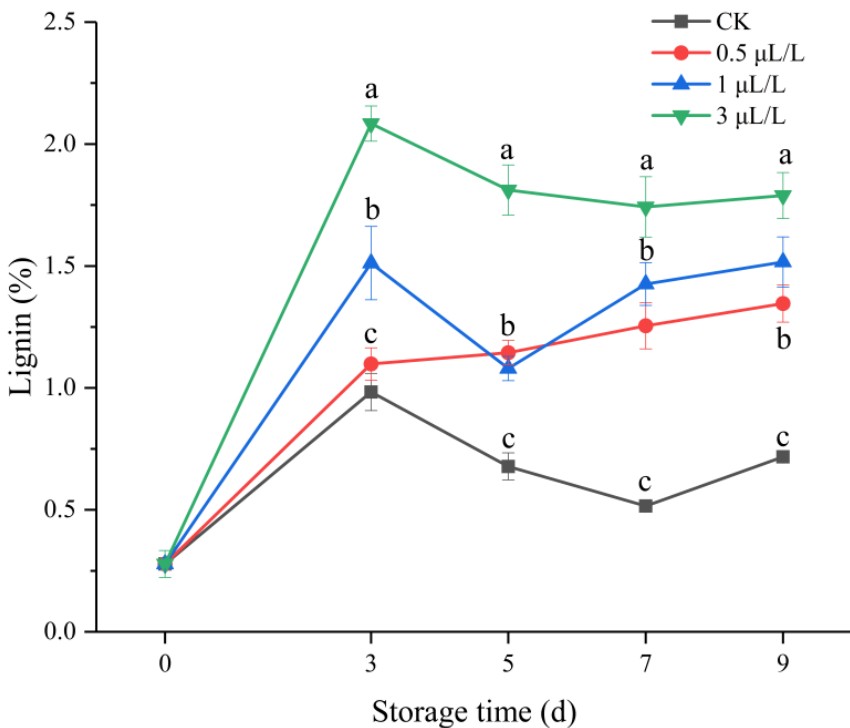

**Figure 5.** Effect of 1-MCP treatment on the lignin content of blueberries during shelf life at 25 ± 1 °C. Data are the means of three replicates ± standard deviation. Values with different letters are significantly different according to Duncan's test (*p* < 0.05).

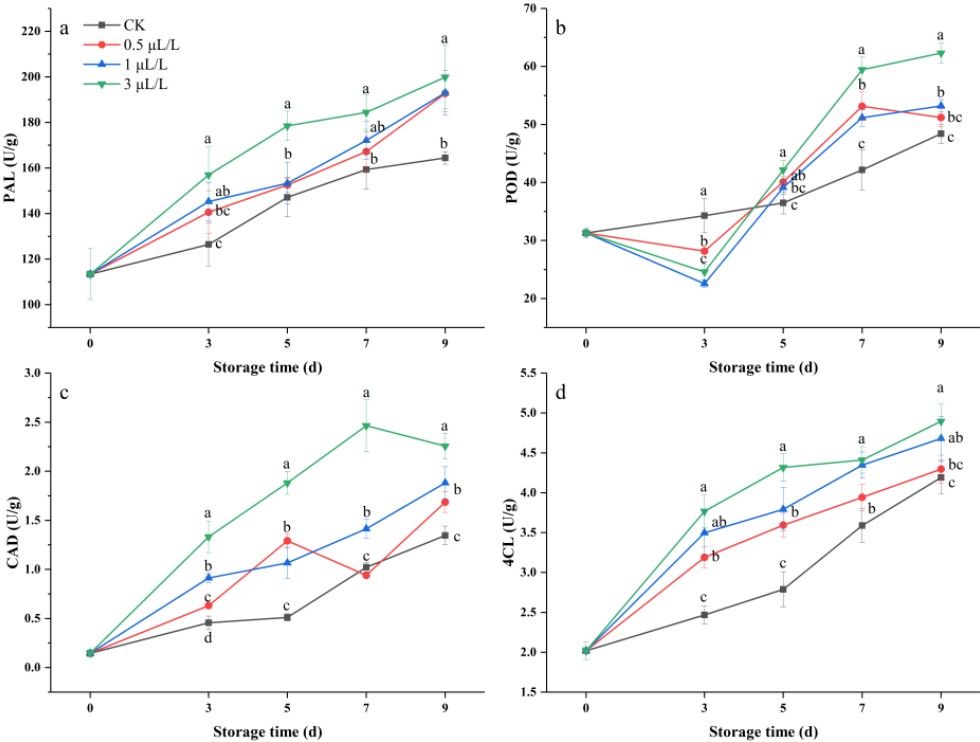

**Figure 6.** Effect of 1-MCP treatment on the PAL activity (**a**), POD activity (**b**), CAD activity (**c**) and 4CL activity (**d**) in blueberries during shelf life at 25 ± 1 °C. Data are the means of three replicates ± standard deviation. Values with different letters are significantly different according to Duncan's test (*p* < 0.05).

The foregoing results indicated that the lignin content was increased by 1-MAP treatment accompanied by activated PAL, POD, CAD and 4CL activity.

### 3.4.3. Cellulose and Hemicellulose Content

As revealed in Figure 7a, cellulose content rapidly decreased from 3 d to 5 d regardless of 1-MCP treatment. However, there was a slower decrease for 1-MCP-treated blueberries; cellulose content was 1.43, 1.93 and 1.90 times more than in untreated blueberries at 5 d ($p < 0.05$), respectively. After 5 d, the cellulose content slightly decreased until the end of the storage time both in 0.5 µL/L 1-MCP-treated and untreated blueberries, whereas in 1 µL/L 1-MCP-treated and 3 µL/L 1-MCP-treated blueberries, it increased as the storage time went on.

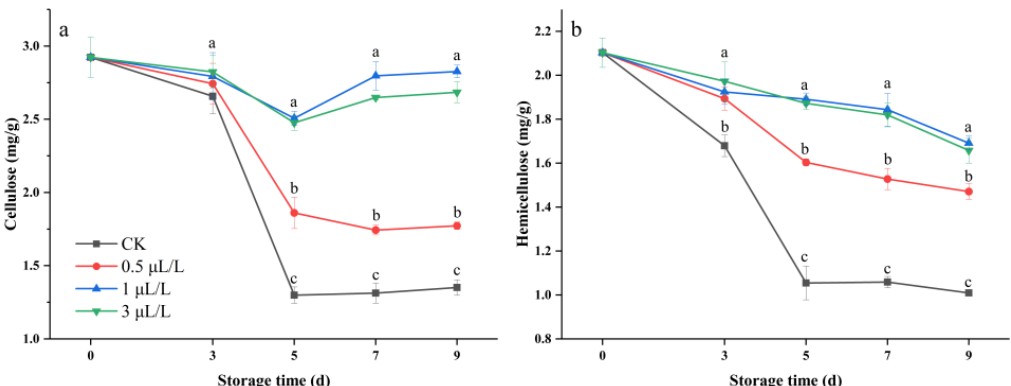

**Figure 7.** Effect of 1-MCP treatment on cellulose (**a**) and hemicellulose content (**b**) of blueberries during shelf life at 25 ± 1 °C. Data are the means of three replicates ± standard deviation. Values with different letters are significantly different according to Duncan's test ($p < 0.05$).

From Figure 7b, the change in hemicellulose content was similar to that of cellulose, which decreased over the entire storage time, regardless of 1-MCP treatment. However, hemicellulose content was significantly higher in 0.5 µL/L 1-MCP-treated, 1 µL/L 1-MCP-treated and 3 µL/L 1-MCP-treated blueberries than in untreated ones from 3 d to 9 d ($p < 0.05$), and 1.45, 1.67 and 1.64 times higher, respectively, than in untreated blueberries.

### 3.5. ATP, ADP, and AMP Contents and Energy Charge (EC)

Changes in the levels of ATP, ADP, AMP and EC in untreated and 1-MCP-treated blueberry are shown in Figure 8. In untreated blueberries, the ATP level and EC decreased as the storage time went on, but the level was significantly higher for 1 µL/L 1-MCP-treated and 3 µL/L 1-MCP-treated blueberries than for untreated ones from 3 d to 9 d ($p < 0.05$). In contrast, the AMP content was increased over the whole storage time, regardless of 1-MCP treatment (Figure 8c). However, in 3 µL/L 1-MCP-treated blueberries, AMP content was significantly lower than in untreated fruit by 22.97%, 31.61%, 39.35% and 46.53% at 3 d, 5 d, 7 d and 9 d, respectively. In contrast, the ADP content was increased at first and then decreased in all blueberries (Figure 8b). Obviously, the ADP content was significantly lower for 3 µL/L 1-MCP-treated blueberries than for untreated ones from 5 d to 9 d (3 µL/L 1-MCP-treated/untreated: 6.04/8.11 at 5 d, 5.63/9.09 at 7 d, 5.72/6.67 at 9 d).

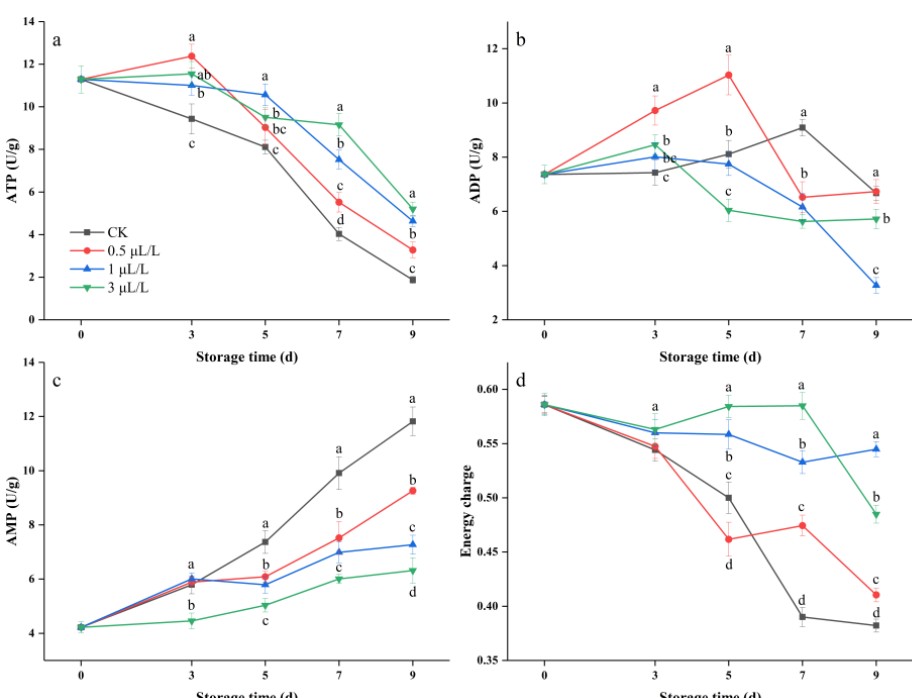

**Figure 8.** Effect of 1-MCP treatment on the ATP (**a**), ADP (**b**) and AMP (**c**) content and EC (**d**) in blueberries during shelf life at 25 ± 1 °C. Data are the means of three replicates ± standard deviation. Values with different letters are significantly different according to Duncan's test (*p* < 0.05).

## 4. Discussion

### 4.1. Effect of 1-MCP Treatment on Blueberry Weight Loss, Decay Incidence and Respiration Rate

Blueberries are quite perishable during storage, and this is accompanied by a deterioration in quality and quantity. As an ethylene receptor inhibitor, 1-MCP is crucial in inhibiting fruit respiration and decay [8,26]. In this work, the weight loss was lower in fruits treated with 1 μL/L and 3 μL/L 1-MCP than in untreated fruits. In contrast, the highest weight loss occurred in the 0.5 μL/L1-MCP-treated blueberries (Figure 1a). Similarly, Grozeff et al. found that 1 μL/L 1-MCP treatment was not effective in reducing weight loss for 'Misty' and 'Blue Cuinex' blueberries, but was effective for 'Blue Chip' blueberries [15]. We speculated that this phenomenon was due to different cultivars having different sensitivity to 1-MCP; which is to say, different concentrations of 1-MCP might have opposite effects on weight loss.

During the postharvest stage, the weight loss and decay incidence increased simultaneously. Generally, a higher respiration rate is particularly important for fruit weight loss and decay. Ji et al. reported that ethanol fumigation treatment effectively delayed the increase in weight loss rate and decay incidence of blueberries simultaneously. As the weight loss rate increases, the decay becomes more severe [24]. The effect of 1 μL/L and 3 μL/L 1-MCP treatment on the respiration rate and decay incidence was obvious, but the effect of 0.5 μL/L 1-MCP treatment was not. Similar results were reported in previous studies on French prunes treated with 1.2 μL/L 1-MCP and apples treated with 1 μL/L 1-MCP [27,28]. In addition, for 0.5 μL/L 1-MCP-treated blueberries, we found that the change in respiration rate was similar to the weight loss. Therefore, the highest weight loss and higher decay incidence in 0.5 μL/L 1-MCP-treated blueberries were ascribed to a higher respiration rate.

### 4.2. Effect of 1-MCP Treatment on Blueberry Firmness, SSC, TA, BAR, Anthocyanins and Sensory Evaluation

Treatment with 1-MCP has delayed the decrease in firmness [15], inhibited SSC and maintained the TA content [8] of the blueberry fruit. In the present study, the firmness of

postharvest blueberry fruit decreased gradually with shelf life ($25 \pm 1\ ^{\circ}C$) time. However, 3 µL/L 1-MCP treatment slowed down the decline in firmness (Figure 2a), accompanied by significant effects on other quality traits, such as SSC, TA, BAR and anthocyanins. Similar effects were also observed in previous studies on blueberry [16], kiwifruit [29] and apple [30]. In contrast, Nesmith et al. (2011) found that 1-MCP accelerated the rate of firmness loss in three different cultivars [31]. It seems contradictory, but that is not the case. Different cultivars of blueberry had very different metabolic mechanisms, e.g., the metabolic mechanism of 'Britewell' blueberry has a special structure.

On the other hand, the results of the sensory evaluation supported the reliability of the quality parameters to a certain extent (Table 1). After 1-MCP treatment, although the values of "taste" and "aroma" decreased, there was no difference between treated and untreated ones. Particularly, the "firmness" and "appearance" values were higher for 3 µL/L 1-MCP-treated than untreated ones at 5 d. In fact, the softening of postharvest blueberries was effectively delayed with 3 µL/L 1-MCP treatment (2 h) in this study, and it did not significantly affect the aroma and taste. Compared to the previous studies that applied 1 µL/L 1-MCP for 12 h and 0.3 µL/L 1-MCP for 24 h [15,26], the treatment of the present study was more efficient and thus provided more time for transportation and sales.

### 4.3. Effect of 1-MCP Treatment on Cell Wall Metabolism and Enzyme Activity

Cell wall polysaccharides, mainly including pectin, cellulose and hemicellulose, are important components of cell wall structure and play key roles in cell wall metabolism. In addition, modifications in the composition of cell wall polysaccharides have been associated with changes in fruit firmness [6]. The reduction in fruit firmness is mainly due to the decomposition of protopectin and cellulose [32]. With time, the protopectin and cellulose degrade to WSP and hemicellulose, respectively. In the present study, the content of protopectin decreased and WSP increased over the whole storage time. Particularly, 3 µL/L 1-MCP treatment inhibited the decomposition of protopectin, such that the content of protopectin and WSP was significant higher and lower, respectively, than in untreated ones (Figure 5a,b). Accordingly, treatment with 3 µL/L 1-MCP also effectively inhibited the activity of PG and PME (Figure 4c,d). Similarly, the content of cellulose and hemicellulose was higher in 3 µL/L 1-MCP-treated blueberries than in untreated ones over the whole storage time (Figure 7). Moreover, blueberry firmness was significantly higher than the untreated group (Figure 2a). These results were also in accordance with previous reports on blueberry, plum and apple [6,12,13].

To the best of our knowledge, there is little available literature concerning the lignification of blueberries. However, the application of 1-MCP has been reported to increase the risk of lignification in kiwifruit [20]. In contrast, other studies have demonstrated that lignification in bamboo shoots and loquat has been reduced by the application of 1-MCP. [33,34]. Hence, the effect of 1-MCP on lignification is different in different varieties. In the present study, 3 µL/L 1-MCP treatment increased the content of lignin and the activities of PAL, POD, CAD and 4CL (Figures 5 and 6); thus, lignification was more serious than in untreated blueberries. However, no negative effects of lignification on food quality were found in the sensory evaluation results (Table 1). In fact, the content of lignin was correlated with fruit firmness [35]. The results of this study indicated that higher lignin content could be responsible for maintaining higher blueberry firmness in the 1-MCP treatment groups.

### 4.4. Effect of 1-MCP Treatment on ATP, ADP and AMP Content and Energy Charge (EC)

Adequate cellular energy supply is an important factor in the quality and storability of fruit [36]. In ETC respiration, ATP is decomposed to ADP by ATPases and produces free phosphate ions to coincide with energy release [37]. In the present study, 1-MCP-treated blueberries showed higher ATP and EC levels than untreated ones (Figure 8a,d). In contrast, the ADP and AMP levels were inhibited by 1-MCP treatment (Figure 8b,c); this effect was particularly obvious with 3 µL/L 1-MCP treatment. Thus, we suggested that the higher quality of the 3 µL/L 1-MCP-treated blueberries is due to a sufficient supply of intracellular

energy. In previous studies, Cheng et al. (2015) reported that 1-MCP treatment alleviated chilling injury and maintained postharvest quality by regulating energy metabolism in 'Nanguo' pears [26]. Similar results had also been reported in kiwifruit and tomato [38,39]: fruits with better quality were the result of adequate mitochondrial energy supply.

## 5. Conclusions

This study explores a short duration and efficient treatment technique for 1-MCP in blueberry. These results shed light on the valuable efficiency of short-term 1-MCP treatment (3 μL/L, 2 h) for reducing the softening of blueberry fruit during shelf life. The 3 μL/L 1-MCP treatment maintained better shelf quality and delayed softening, which was related to higher firmness and sufficient energy supply, as well as inhibition of the activity of cell-wall degradation, including PG and PME. In addition, the lignin content was increased by 3 μL/L 1-MCP treatment because the activities of PAL, POD, CAD and 4CL were induced. The results suggested that short-term 1-MCP treatment was an effective technique for delaying the postharvest softening process and extending the shelf life of blueberry fruit. As a result, there has been an improvement in production efficiency.

**Author Contributions:** Conceptualization and methodology: H.Y. and R.W.; investigation: H.Y., R.W., N.J., L.B., G.W. and X.L.; formal analysis: H.Y., J.L. (Jiangkuo Li) and J.L. (Jiqing Lei); data curation: H.Y. and C.M.; writing—original draft preparation: H.Y.; writing—review and editing: H.Y. and R.W. All authors have read and agreed to the published version of the manuscript.

**Funding:** This research was funded by the Guizhou Province Key Technology Research and Development and Application of Innovation Base for Agricultural Products Primary Processing (Qi keZhong Yin Di (2020) 4018) and the Discipline and Master's Site Construction Project of Guiyang University by Guiyang City Financial Support Guiyang University (SY-2020).

**Data Availability Statement:** Not applicable.

**Conflicts of Interest:** The authors declare no conflict of interest.

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
