# Peer review of "Regulation of Cell Wall Degradation and Energy Metabolism for Maintaining Shelf Quality of Blueberry by Short-Term 1-Methylcyclopropene Treatment"

_agronomy, doi:10.3390/agronomy13010046_

Round 1

Reviewer 1 Report

The manuscript titled “Short-term 1-methylcyclopropene treatment maintenance of shelf quality of blueberry associated with regulation of cell wall degradation and energy metabolism” is a great paper. The manuscript deals with an important question of the blueberry post-harvest.

Please describe the aim of the research, which is missing from the abstract.

Ripening stage of the blueberries is important in this research; therefore it would be useful to have some more information about it.

To point 2.7. Were the participants treated or untreated? Did you have any experience about blueberry evaluation? Is it possible to add their number (how many judges made the sensory evaluation?), age in age groups, and sex? Were the participants in the sensory evaluation give points from 1 to 10, where 10 the best was?

Author Response

Response to Reviewer 1 Comments

Point 1: To point 2.7. Were the participants treated or untreated? Did you have any experience about blueberry evaluation? Is it possible to add their number (how many judges made the sensory evaluation?), age in age groups, and sex? Were the participants in the sensory evaluation give points from 1 to 10, where 10 the best was?

Response : Thanks for your suggestion. We have studied blueberry preservation technology for 11 years and have experience with sensory evaluation of blueberries. The problem with the “point 2.7” of the manuscript is that it was improperly written and it has been modified carefully in the revised manuscript.

In the previous manuscript, line 188-190 on page 4:

Sensory evaluation was conducted using a 9-point hedonic scale. The participants were asked to evaluate each sample for the sensory attributes of appearance, color, flavor and taste. For scoring criteria, we referred to Nirmal et al. (2020).

was change to revised manuscript, line 215-222 on page 5:

Sensory evaluation was conducted using a 9-point hedonic scale and performed in a sensory laboratory at Guiyang University. Ten trained and regular blueberry consumers (there are five men and five women, ranging in age from 20 to 56)The participants were asked to evaluate each sample for the sensory attributes of appearance, color, flavor and taste. For scoring criteria, we referred to Nirmal et al. (2020). Each attribute was scored as follows: 9 = Like extremely; 8 = Like very much; 7 = Like; 6 = Like slightly; 5 = Neither like nor dislike; 4 = Dislike slightly; 3 = Dislike moderately; 2 = Dislike; 1 = Dislike extremely.

Reviewer 2 Report

The manuscript entitled “Short-term 1-methylcyclopropene treatment maintenance of shelf quality of blueberry associated with regulation of cell 3 wall degradation and energy metabolism” evaluates the response of different concentrations of 1 MCP on the shelf life and quality of blueberries.  The manuscript differs when compared to other studies on 1-MCP further lignin, enzyme activities and cell wall metabolism, making it more interesting. The manuscript has a lot of conceptual and grammatical mistakes that seriously need to be addressed for its suitability. In some instances, the intext referencing is not proper, fix that as well.

I would advise the authors to revise the title of this manuscript.

1. Abstract:

The abstract requires major revision, there are lots of errors in it:

For example:

Line 18: In compared to other

Line 20: the SSC and BAR were reduced compared to those untreated,

Line 20, Line 22, Line 24, and Line 26

Even line 12

In line 32: What is the semicolon doing after the last key work?

2. Introduction

The introduction is informative; however, the English is not good and requires editing

For example:

Lines 42 and 43 need to be revised

Line 44: Blueberry is rich of phenolic acids???? Revise this

There are some other lines that have grammatical errors, please check throughout this section.

Objective should be aligned according to the special issue theme.

3. Material and method

Line 91: This is not the first time you are mentioning blueberry, why did you decide to include its scientific name in this section???

In the same line: “were picked from an seven-year-old” there is a grammatical error here

Line 92: what is PR???

Line 93: same maturity stage

Why was the treatment applied for 2hrs????? and how was it applied??? Dipping or spraying??? Did you apply the treatment as a coating????

In section 2.1 there is no full stop at the end of the section heading while in section 2.2 there is one, please pay attention to details.

Line 107: was calculated not measured

Line 119: replace via by using

Line 120: “BAR was calculated from SSC and TA” how??? Specify

Line 123: You did not homogenize the samples before centrifugation?

The way you expressed the speed of a centrifuge in line 118 is not the same as in line 124, why? Be consistent.

Line 134: what do you mean by cold ethanol?? Was it from the fridge or stored in a very cold environment?

Revise line 135-136

In section 2.5.2, check the way you wrote °C and check throughout the document

Section 2.6 is supposed to be one paragraph

In terms of the sensory evaluation, were the participants trained? Or they were randomly asked to participate in the study without any training?

Overall, enough parameters were analyzed but the section requires English editing to make it reads well.

 Results

In Figure 1c it is clear whether the lines have error bars or not, while in the other figures no error bars are visible

Again, in Figure 2, the error bars are not clear

In figure 2d, the numbers on the x-axis are not in line with the points in the lines, why???

It is not necessary to put a dot after the word ‘Table’, line281 and 289

Line 290: Is it date or data????

The first letter of the parameters measured should be capitalized in all your figures or graphs

Line 313: there must be a space between L and 1

Line 342: there must be a space between the bracket and d…..in the same line you started a new line with a small letter

Line 360: cellulose slightly increased from d 5 to 9 on the untreated blueberries

Line 348: is it MAP or MCP?

Overall, the results are well described but some lines are not clear in some figures and the error bars are not visible enough, please check this.

Discussion

There is no need for this line “In present study, the respiration rate, weight loss and decay incidence occurred 401 together (Fig. 1 a-c)” in line 401-402, same as 403 and 403

Paragraph 400-408 needs to be revised, it lacks science

Minimize repeating your results in this section
